# Development of a Duplex Serological Multiplex Assay for the Simultaneous Detection of Epstein-Barr Virus IgA and IgG Antibodies in Nasopharyngeal Carcinoma Patients

**DOI:** 10.3390/cancers15092578

**Published:** 2023-04-30

**Authors:** Jennifer Schieber, Miranda Pring, Andy Ness, Zhiwei Liu, Wan-Lun Hsu, Nicole Brenner, Julia Butt, Tim Waterboer, Julia Simon

**Affiliations:** 1Division of Infections and Cancer Epidemiology, German Cancer Research Center (DFKZ), 69120 Heidelberg, Germany; jenny_schieber@gmx.de (J.S.);; 2Faculty of Biosciences, Heidelberg University, 69120 Heidelberg, Germany; 3Bristol Dental School, University of Bristol, Bristol BS8 1QU, UK; 4Division of Cancer Epidemiology and Genetics, National Cancer Institute, Bethesda, MD 20892, USA; 5Data Science Center, College of Medicine, Fu Jen Catholic University, New Taipei City 242062, Taiwan; 6Master Program of Big Data in Biomedicine, College of Medicine, Fu Jen Catholic University, New Taipei City 242062, Taiwan

**Keywords:** antibodies, EBV, IgA, IgG, Intelliflex, multiplex serology, NPC

## Abstract

**Simple Summary:**

IgA and IgG antibodies against Epstein-Barr virus (EBV) proteins in human serum are well-known markers for EBV-positive nasopharyngeal carcinoma (NPC). Bead-based multiplex serology assays can characterize antibodies against several antigens simultaneously; however, these assays were, so far, specific for single antibody isotypes. Here, we describe the development of a duplex serological multiplex assay for the simultaneous detection of EBV IgA and IgG antibodies in a combined assay. The novel duplex assay decreases costs and effort for future epidemiological studies in EBV and NPC research and could serve as a model for other duplex multiplex applications.

**Abstract:**

Epstein-Barr virus (EBV) IgA and IgG antibodies in serum from nasopharyngeal carcinoma (NPC) patients are well-established markers for EBV-positive NPC. Luminex-based multiplex serology can analyze antibodies to multiple antigens simultaneously; however, the detection of both IgA and IgG antibodies requires separate measurements. Here we describe the development and validation of a novel duplex multiplex serology assay, which can analyze IgA and IgG antibodies against several antigens simultaneously. Secondary antibody/dye combinations, as well as serum dilution factors, were optimized, and 98 NPC cases matched to 142 controls from the Head and Neck 5000 study (HN5000) were assessed and compared to data previously generated in separate IgA and IgG multiplex assays. EBER in situ hybridization (EBER-ISH) data available for 41 tumors was used to calibrate antigen-specific cut-offs using receiver operating characteristic (ROC) analysis with a prespecified specificity of ≥90%. A directly R-Phycoerythrin-labeled IgG antibody in combination with a biotinylated IgA antibody and streptavidin-BV421 reporter conjugate was able to quantify both IgA and IgG antibodies in a duplex reaction in a 1:1000 serum dilution. The combined assessment of IgA and IgG antibodies in NPC cases and controls from the HN5000 study yielded similar sensitivities as the separate IgA and IgG multiplex assays (all > 90%), and the duplex serological multiplex assay was able to unequivocally define the EBV-positive NPC cases (AUC = 1). In conclusion, the simultaneous detection of IgA and IgG antibodies provides an alternative for the separate IgA/IgG antibody quantification and may present a promising approach for larger NPC screening studies in NPC endemic areas.

## 1. Introduction

Nasopharyngeal carcinoma (NPC) is a head and neck cancer closely associated with Epstein-Barr virus (EBV) infection [1]. Approximately 80% of all new NPC cases in 2012 were diagnosed in Southern China and Southeast Asia, where NPC is endemic [2]. Due to the close association of EBV infection with NPC in endemic areas, EBV serum antibodies can be used as diagnostic markers [3]. Moreover, the defined population at risk allows EBV antibodies to be used for early NPC detection and NPC screening [4]. 

In NPC non-endemic regions, which include Europe and the US, NPC is very rare [2], and only 60–76% of all NPCs are EBV positive [5,6,7,8]. The detection of EBV RNA in NPC tumor tissue is the gold standard for determining the EBV status of NPC [9]. EBV-positive NPCs from non-endemic areas show a similar histological and serological pattern as NPC from NPC endemic areas, which are mostly from non-keratinizing histological types [1,10]. 

The first approaches to detecting EBV serum antibodies in NPC patients were based on indirect immunofluorescence [11,12,13], which were later replaced by ELISA-based methods. A combined VCA/EBNA1 IgA ELISA has been shown to be the best marker and the preferred screening method for NPC within Southern China [14]. However, the false positive rate of this serological screening method is relatively high [15]. 

In 2018, a novel microarray-based approach led to the development of a highly sensitive and specific EBV antibody panel for the diagnosis of NPC, namely the EBV antibody risk stratification signature [16]. The antibody signature combines 13 EBV IgA and IgG antibodies and outperforms the ELISA-based markers in terms of accuracy. The novel EBV antibody markers were adapted to multiplex serology [17], a high-throughput serological assay, and validated in studies within NPC endemic and non-endemic areas [10,18]. The promising novel screening approach is currently investigated in large prospective studies from NPC-endemic regions.

While multiplex serology [17] can be used to characterize antibodies against up to 42 antigens simultaneously in one well, every secondary antibody isotype, i.e., IgA or IgG, has to be quantified separately. The aim of our study was to develop a quantitative Duplex multiplex serology assay, which can characterize IgA and IgG antibodies against multiple antigens in one reaction. This is challenging due to the different titers of IgA and IgG antibodies in serum and the non-availability of a second reporter channel in the Luminex 200 analyzer. Here, we introduce a quantitative Duplex multiplex assay based on the recently developed xMAP Intelliflex system. The Intelliflex is a flow-based multiplex platform which, in contrast to the Luminex 200 analyzer, has the ability to analyze two parameters, e.g., two secondary antibody isotypes, for each analyte. To the best of our knowledge, this system represents the first bead-based multiplex platform capable of the simultaneous detection of two parameters per analyte. This is based on a dual reporter system, providing two reporter lasers in addition to one classification laser for the classification of the beads. The reporter laser 1 (RP1) excitation is suited for 532 nm, and the reporter laser 2 (RP2) excitation for 405 nm. The emission spectra range from 565 to 585 nm (red-orange light) for RP1, and from 421 to 441 nm for RP2 (violet light). 

The presence of the two reporter channels for analyte detection in the multiplex setting allows the simultaneous quantitative detection of IgA and IgG antibodies against several EBV antigens, which can save time, effort, and precious serum samples, and thus improve the cost-effectiveness for NPC screening in endemic areas. Here, we introduce the development of a Duplex assay for the simultaneous detection of IgA and IgG antibodies by choosing appropriate pairs of secondary antibody/dye combinations and determining the appropriate serum dilution. We validated the novel system by comparing an NPC study previously published as a Singleplex assay with the newly developed Intelliflex Duplex assay. 

## 2. Materials and Methods

### 2.1. Study Material/Study Population

For assay development, pooled serum specimens of people known to have high EBV IgA antibody titers (e.g., NPC patients) and pooled serum specimens of general-population controls were used. The samples were derived from several studies in Taiwan and were pooled to assess the differences in EBV serology among several laboratories in a previous study [19]. 

For the final evaluation of the Duplex assay, sera from the Head and Neck 5000 clinical cohort study were reanalyzed. The same sera, including 98 incident nasopharyngeal carcinoma cases and 142 matched laryngeal squamous cell carcinomas (LSCC), have been analyzed for their EBV IgA and IgG antibodies in a Singleplex assay using a Luminex 200 analyzer [10]. Briefly, the Head and Neck 5000 study is one of the largest clinical cohort studies of patients with head and neck cancer, with 5511 participants recruited in 76 centers within the United Kingdom. A detailed study description is available elsewhere [20]. 

### 2.2. Serology

#### 2.2.1. Singleplex Assay

Serological testing of antibodies was performed with multiplex serology, a high-throughput assay allowing the detection of antibodies against multiple antigens at the same time [17]. MagPlex Microspheres (Luminex Corp., Austin, TX, USA), also called beads, including two fluorescent dyes of different ratios for exact identification, were derivatized with glutathione-casein, allowing the binding of glutathione-S-transferase (GST)-tagged antigens. For assay development experiments, antigens of five common general infection markers of EBV (EBNA1, EBNA1 peptide (pep), ZEBRA, VCAp18, and EA-D), which were previously validated for multiplex serology [21], were used. EBNA1 displays a truncated version consisting of 317 amino acids of the 641 amino acid-long original EBNA1 protein, while EBNA1 pep covers a peptide sequence of 36 amino acids. For the final evaluation of NPC and LSCC sera from the Head and Neck 5000 study, 13 NPC-specific antigens were included, which were derived from an antibody risk stratification signature (EBNA1 (BKRF1), EBNA1 peptide (pep) (BKRF1), ZEBRA (BZLF1), VCAp18 (BFRF3), EA-D (BMRF1), BXLF1, LF2, BZLF1, BORF1, BFRF1, BGLF2, BRLF1, BPLF1) and have been validated previously [18]. 

Sera were pre-incubated in a dilution twice as concentrated as the desired final dilution in a pre-incubation buffer consisting of phosphate-buffered saline (PBS) containing 2 mg/mL casein, 2 g/L GST-tag lysate, 5 g/L polyvinyl alcohol, and 8 g/L polyvinyl-pyrrolidone [22]. Subsequently, pre-incubated sera were mixed with equal volumes of antigen-coupled beads. IgA and IgG antibodies were measured separately in a final serum dilution of 1:100 and 1:10,000, respectively. For detection of the bound IgA and IgG serum antibodies, either biotinylated anti-IgA (1:1000) plus Strep-PE (1:750) or biotinylated anti-IgG (1:1000) plus Strep-PE (1:750) were used, respectively (Table 1). 

#### 2.2.2. Development of a Duplex Assay

For the Duplex assay, antibodies against the same 13 NPC-specific antigens as described for the Singleplex assay were evaluated. These GST-tagged antigens bind to the glutathione-casein-coupled MagPlex microspheres, allowing IgA and IgG antibodies from serum specimens to bind. In contrast to the Singleplex assay, IgA and IgG antibodies were measured simultaneously in one assay (Duplex assay), which reduces the assay reagents and effort, but requires a common serum dilution. Furthermore, the optimal combination of secondary antibodies and Strep-conjugates, as well as their concentration, had to be determined during the assay development. 

For initial assay development experiments, IgA and IgG from human serum (Sigma I2511 and I4036) were used, which were available in large quantities, contrary to precious patient’s sera. IgA and IgG from human serum are reactive for the EBV general infection markers, and comparable to antibodies from serum specimens of people known to have high EBV antibody titers. The assay principle is similar in this case, i.e., the antibodies were bound indirectly to the glutathione-coupled beads via GST-tagged EBV antigens. First, different conditions to detect IgA and IgG were tested. For this, various directly labeled or biotinylated secondary antibodies (Table 1) were serially diluted from 1:50 to 1:6400. Second, the Strep-dye-conjugates were serially diluted from 1:100 to 1:102,400 for determination of the optimal dye concentration for the Duplex assay. 

To find the optimal serum dilution for the simultaneous detection of IgA and IgG antibodies, pooled serum specimens of people known to have high EBV IgA antibody titers (e.g., NPC patients) and pooled serum specimens of general-population controls from several studies in Taiwan were titrated in 1:3 steps from 1:100 to 1:218,700. IgA and IgG were detected with the final combination of secondary antibodies, anti-IgG-PE (1:500) and anti-IgA-Biotin (1:800)/Strep-BV421 (1:500), in the same reaction. 

In the final assay, sera from the Head and Neck 5000 clinical cohort study were pre-incubated as described above in a 1:500 dilution for a final dilution of 1:1000. Of the 240 sera measured in the Singleplex assay, 236 sera were available for characterization of IgA and IgG levels in the Duplex assay format. The IgA and IgG antibodies were detected with a directly coupled anti-IgG-PE antibody (1:500) and, in the same reaction, a biotinylated anti-IgA antibody (1:800). Subsequently, the bead suspension was incubated with Strep-BV421 (1:500) or Strep-SB436 (1:800) for detection of IgA antibodies (Table 1). 

### 2.3. Molecular Tumor Analysis

Epstein-Barr virus small RNA 1 (EBER-1) in situ hybridization (EBER-ISH) was used to determine the EBV status of NPC cases. EBER-ISH was available for a total of 41 NPC cases, either derived from pathology reports or from sectioning and staining of available formalin-fixed paraffin-embedded (FFPE) tumor blocks. EBER-ISH analysis for the 41 NPC cases has been described in detail [10]. 

### 2.4. Statistical Analysis

Statistical analysis was performed using GraphPad Prism 9.0 (GraphPad Software, Inc., La Jolla, CA, USA) or SAS enterprise guide 7.1 (SAS Institute, Cary, NC, USA).

Antigen-specific cut-offs were determined for all IgA and IgG antibodies using receiver operating characteristic (ROC) analysis of cases with available EBER-ISH status (n = 41) and a prespecified specificity of ≥90%. All resulting cut-off values for the Luminex Singleplex assay and the SB436-Duplex and BV421-Duplex assays, and resulting sensitivities and specificities are listed in Appendix A. A technical cut-off of 30 MFI, referring to the lower limit of quantitation, was applied to all samples with calculated cut-offs below 30 MFI.

The diagnostic accuracy of a panel consisting of 13 antibodies (antibody risk stratification signature) [16] was assessed by calculating the AUC on dichotomized values.

## 3. Results

### 3.1. Assay Development

#### 3.1.1. Selection of Antibody-Dye Combinations

Several anti-IgA and anti-IgG antibodies that were either biotinylated or directly coupled to fluorescent dyes (Table 1) were tested in several dilutions against antigen-bound IgA and IgG antibodies from human serum to determine the optimal dilution factor and verify the antibody specificity. All tested antibodies were originally derived from the same goat anti-human secondary antibody, which has been previously used for IgA and IgG multiplex serology assays [10]. Overall, biotinylated antibodies in combination with a Streptavidin (Strep)-dye conjugate resulted in higher MFI values compared to directly dye-labeled antibodies (Appendix A). The weakest signal was observed for directly labeled Dylight405 antibodies, which were thus not further considered for the assay. Streptavidin-conjugated violet dyes Brilliant Violet 421 (Strep-BV421) and Super Bright 436 (Strep-SB436) gave much stronger signals than Dylight405-labeled antibodies but were not as bright as the Streptavidin-conjugated phycoerythrin (Strep-PE) dye. Directly PE-labeled antibodies were less bright than the combination of a biotinylated antibody with Strep-conjugated PE. 

Signals for IgG antibodies were generally stronger compared to IgA antibodies. Therefore, a biotinylated IgA secondary antibody in combination with a violet dye (Strep-BV421/Strep-SB436) was selected to allow for signal amplification and combined with a directly PE-coupled anti-IgG antibody. Compared to previously used biotinylated IgA/IgG antibodies used with Strep-PE conjugates, the signals of both the anti-IgA with Strep-conjugated BV421 or SB436 and the directly PE-labeled anti-IgG antibodies were lower. 

The final choice of antibody dye combinations was tested in several dilutions and is shown in Figure 1. The anti-IgA antibody is shown in combination with Strep-BV421, which can be replaced by Strep-SB436. All tested antibodies were isotype specific, which was demonstrated by the absence of responses of the anti-IgA and anti-IgG antibodies when tested against IgG and IgA from human sera, respectively. A final antibody dilution of 1:500 and 1:800 for anti-IgG and anti-IgA antibodies, respectively, was chosen for further assay development. 

#### 3.1.2. Determination of the Optimal Serum Dilution Factor for the Simultaneous Detection of IgA and IgG Antibodies

The simultaneous quantitative detection of IgA and IgG antibodies is a challenge, as IgA antibodies are usually quantified in a 1:100 dilution, while IgG antibodies require a 1:1000 or even 1:10,000 serum dilution [18]. Thus, the optimal serum dilution factor for a simultaneous IgA and IgG measurement was to be determined with the previously chosen antibody/dye combination.

Pooled sera from NPC patients and healthy controls were serially diluted, and anti-EBV IgA and IgG antibodies were quantified separately with the chosen secondary antibody/dye combination. Anti-IgA and anti-IgG antibodies against five EBV general infection markers (EBNA1, EBNA1pep, VCAp18, EA-D, and ZEBRA) were detected. Antibody responses against two of the five antigens (VCAp18 and EBNA1) are displayed in Figure 2. Both IgG and IgA antibody levels were higher in sera from NPC patients compared to sera from controls (Figure 2). Comparing IgA/IgG signal intensities, IgG antibodies were more abundant than IgA antibodies in both NPC patients and controls. While IgG antibodies against the EBV antigens were observed in both NPC cases and controls, IgA antibodies were barely detectable in the control sera.

To determine the optimal serum dilution for the simultaneous detection of IgG and IgA antibodies, two factors must be considered. First, IgA antibodies cannot be detected in very high serum dilutions. Second, at very high serum concentrations, the antibodies can form complexes resulting in a pro-zone effect, i.e., lower MFI values at the highest concentrations (Figure 2C). Thus, for the simultaneous detection of IgA and IgG antibodies, a serum dilution of 1:1000 was chosen since both IgA and IgG levels were measurable in a quantitative manner with this serum dilution, especially in sera of NPC patients. 

#### 3.1.3. Duplexing of IgA and IgG Antibodies

To exclude the possibility of undesirable interactions between the secondary antibodies and Strep-conjugates used for detection of the IgA and IgG antibodies present in the patient’s serum, separate IgA and IgG detection assays were compared to the Duplex assay, where IgA and IgG antibodies are detected simultaneously. Therefore, IgA and IgG antibodies in serially diluted pools of NPC sera and control sera of healthy individuals were detected, either in separate reactions (Singleplex) or in the same reaction (Duplex).

For the biotinylated anti-IgA antibody, two different violet dyes (Strep-BV421 and Strep-SB436) were chosen because of their bright signal compared to other violet dyes. To compare these two violet dyes directly, IgA and IgG antibodies were quantified in two setups—the anti-IgG-PE in combination with anti-IgA-Biotin and Strep-BV421 (PE/BV421 Duplex), and the anti-IgG-PE in combination with anti-IgA-Biotin and Strep-SB436 (PE/SB436 Duplex). Both the PE/BV421 and the PE/SB436 combination were measured either separately in a Singleplex assay or in the same reaction (Duplex assay). 

For the PE/BV421 Duplex, two independent experiments were conducted. In both experiments, IgG signals in the Duplex were almost as high as in the Singleplex assay, with a good correlation coefficient (R^2^ = 0.88–0.93) and linear regression slopes of 0.90–1.00, indicating low systematic variation. In contrast to this, the IgA signals were slightly reduced when measured in the same reaction with IgG compared to measuring IgA alone. Here, the correlation was also very high (R^2^ = 0.90–0.94), but the linear regression slopes varied between 0.68 and 0.83. Thus, when detecting IgA and IgG simultaneously, lower signals for IgA might be detected in the PE/BV421 Duplex. 

For the PE/SB436 Duplex, two independent experiments showed that IgG signals are constant when measured in the Duplex compared to separate measurements, with a high correlation (R^2^ = 0.93–0.97) and linear regression slopes of 0.99–1.07, indicating that the IgG detection is robust even when measured together with IgA. However, in contrast to the PE/BV421 Duplex, the IgA signals, in this case, were higher in the Duplex reaction compared to the Singleplex reactions, with linear regression slopes of 1.08 to 1.36, but the Singleplex and Duplex data still showed a very good correlation (R^2^ = 0.98–0.99). 

In conclusion, the detection of IgG signals was found to be very stable when measured together with IgA, whereas IgA signals measured together with IgG slightly deviated from those measured separately in independent experiments. However, in all experiments, the Singleplex data correlated well with the Duplex data.

### 3.2. Study Data: Simultaneous Measurement of EBV IgA and IgG Antibodies in NPC Cases and Controls from the Head and Neck 5000 Study

NPC cases and matched laryngeal squamous cell carcinomas (LSCC) from the Head and Neck 5000 study were characterized for their IgA and IgG antibody levels against 13 EBV antigens in a Duplex assay design. The separate IgA and IgG antibody detection for the same NPC cases and controls has been previously published [10]. Of all 240 sera characterized in the Luminex 200, 236 were available for a repeated assay in Duplex format. The assay was performed twice, as a PE/SB436 Duplex and as a PE/BV421 Duplex assay. 

#### 3.2.1. Correlation of IgA and IgG Antibody Responses in Singleplex (Luminex 200) and Duplex (Intelliflex) Assay 

The newly generated data on IgA and IgG antibody levels from the Duplex assay in a 1:1000 serum dilution was compared to previously generated IgA and IgG Singleplex data from the Luminex 200. In the previous Singleplex assay, IgA antibodies were quantified in a 1:100, and IgG antibodies in a 1:10,000 serum dilution.

For IgG antibodies, the data from both the PE/SB436 Duplex and the PE/BV421 Duplex correlated well with the previously generated IgG data (Figure 3A,B), although the serum dilution differed ten-fold, which is compensated by the biotinylation of the IgG antibody in combination with Strep-PE in the Singleplex assay, allowing significant signal amplification. The coefficients of correlation (R^2^) for the PE/SB436 Duplex and PE/BV421 Duplex IgG data with the previously generated IgG data were 0.86 and 0.88, respectively. The linear regression slopes were 0.97 and 0.95 for the PE/SB436 and PE/BV421 Duplex, respectively, and revealed nearly no difference in signal intensity between the independently performed Singleplex and Duplex assays (Figure 3A,B).

Similarly, IgA data from the PE/SB436 and PE/BV421 Duplex assays were compared to the IgA Singleplex data, which was generated with a 1:100 serum dilution. While the same biotinylated IgA antibody was used in all assays, the Singleplex assay used Strep-PE, and the Duplex assay used either Strep-SB436 or Strep-BV421 as reporter dyes. The R^2^ values for the PE/SB436 and PE/BV421 Duplex IgA levels with the previously generated IgA levels were 0.61 and 0.60, respectively (Figure 3C,D). While low antibody levels could not be assessed in the 1:1000 serum dilution of the Duplex assay, the 1:100 dilution of the Singleplex assay reached the detection limit for some cases with high IgA antibody levels and was not quantitative in the high MFI ranges (Figure 3C,D). 

Comparing the IgA signals detected with Strep-SB436 with the IgA signals detected with Strep-BV421, the correlation was high (R^2^ = 0.95, Figure 3E), and the SB436 dye showed slightly lower signal intensities than the BV421 dye (slope = 0.89, Figure 3E). The IgG-PE signals in the PE/SB436 Duplex correlated well with IgG-PE signals in the PE/BV421 Duplex and did not show any differences in signal intensities (R^2^ = 0.97, slope = 1.01, Figure 3F).

In conclusion, the simultaneous detection of IgA and IgG antibodies does not seem to be affected by the presence of a second reporter antibody/dye. Both violet Strep-conjugates, Strep-BV421 and Strep-SB436, are adequate violet dyes for the detection of IgA antibodies. However, the use of Strep-BV421 results in slightly higher IgA signals due to the higher brightness. The detection of IgG antibodies was not affected by either of the two violet dyes.

#### 3.2.2. Seropositivity by EBER-ISH Status

To further assess whether the changed signal intensities have an impact on the classification of NPC cases and controls, all sera from the Head and Heck 5000 study were analyzed in parallel to the previous analysis [10]. IgA and IgG antibodies against all 13 antigens were compared to EBER-ISH tumor status available for 41 NPC (29 positive and 12 negative). Antigen-specific cut-offs were determined using receiver operating characteristic (ROC) analysis with a minimum specificity of 90% for EBER-ISH negative NPC. The highest sensitivities were observed for LF2 IgG antibodies, with 97% sensitivity at 100% specificity for all three assays (Luminex 200 Singleplex, Strep-BV421 Duplex, and Strep-SB436 Duplex) and BGLF2 IgG antibodies with 97% sensitivity at 92% specificity for the Singleplex assay and 100% sensitivity at 92% specificity for both Duplex assays (Table 2, Figure 4). The lower sensitivity for BGLF2 in the Singleplex assay is based on one case with MFI values close to the cut-off value (Figure 4A). The cut-off values for these antigens barely changed, ranging from 241 MFI for BGLF2 IgG in the Luminex 200, to 251 MFI and 271 MFI in the PE/BV421 and PE/SP436 Duplex assays, respectively (Table 2). The cut-offs and resulting sensitivities and specificities for all antigens and their respective IgA and IgG antibodies are shown in Appendix A. Mostly, although cut-off values differed due to differences in signal intensities, especially for IgA, the sensitivities and specificities were similar, and changes were mostly based on single cases with MFI values close to the cut-offs. Of note, for some IgA antibodies and very few IgG antibodies, the cut-off values had to be raised to 30 MFI, which is considered the lower limit of quantitation, resulting in a lower sensitivity for some antibodies (Appendix A). 

#### 3.2.3. Performance of the 13-Marker Antibody Risk Stratification Signature

To identify whether the slight differences in the sensitivity of some markers influence the performance of the marker panel in the identification of EBER-ISH-positive NPC, the area under the receiver operating characteristic curve (AUC) of a model containing 13 IgA and IgG markers (previously published as antibody risk stratification signature) was calculated based on dichotomized values. All models based on assay data from the Singleplex and the two Duplex assays showed a complete separation of data points (AUC = 1). Furthermore, a previously published four-marker model, including LF2 IgG, LF2 IgA, BGLF2 IgG, and EA-D IgA, was tested in all assays, similarly causing a complete separation of data points, indicating that the Duplex assays perform as well as the Singleplex assay in identifying EBV-positive NPC cases.

## 4. Discussion

The newly developed quantitative Duplex multiplex serology assay presented here allows combining the quantification of IgA and IgG antibodies against several EBV antigens in one assay. This assay is especially useful for large NPC screening studies or trials, and reduces not only effort and costs, but also the volume of precious serum samples. A study of 98 NPC cases and 142 LSCC controls were used to compare the performance of the novel assay with the separate quantification of IgA and IgG antibodies and yielded an equally good performance in defining EBV-positive NPC.

The basis for the Duplex multiplex serology assay is the xMAP Intelliflex system, which provides two reporter channels for analyte detection. The first reporter channel (RP1) has an emission maximum of 565–585 nm and can be used similarly to the reporter channel in the Luminex 200, e.g., with R-phycoerythrin (PE). The Intelliflex system also provides a mode to measure RP1 separately in a Luminex 200 or Flexmap 3D mode. The second reporter channel (RP2) has an emission ranging from 421–441 nm in the violet spectrum. Several violet fluorescence dyes are available, and three dyes, including Dylight 405, Brilliant Violet 421 (BV421), and Super Bright 436 (SB436), have been tested for the presented study. While Dylight 405, which was available as a dye directly coupled to an anti-IgA/IgG antibody, yielded very low signal intensities, our experience with Strep-BV421 and Strep-SB436 conjugates was satisfactory, although signals were roughly 40% lower for the violet dyes in comparison to Strep-PE conjugates regularly used for multiplex serology assays. 

The largest challenge of the assay development was to find a common serum dilution and an appropriate secondary antibody combination for the detection of IgA and IgG antibodies. While IgA antibodies were quantified in a 1:100 dilution and IgG antibodies in a 1:10,000 dilution in previous assays, the requirement for the Duplex multiplex serology assay was a common serum dilution. A final serum dilution of 1:1000 was found to be suitable for the quantification of both IgA and IgG antibodies. However, not only the serum dilution, but also the choice of secondary antibody/dye combinations can influence the signal intensities. While the combination of biotinylated antibodies and Strep-dye conjugates results in signal amplification, this is not valid for directly-labeled secondary antibodies. Thus, the anti-IgG-PE antibody, which was chosen for the Duplex assay, did not yield higher signals in a 1:1000 dilution in the Duplex assay compared to the Singleplex assay, where antibodies were detected with an anti-IgG-Biotin and Strep-PE conjugate in a 1:10,000 serum dilution (Figure 3A,B). Other factors, e.g., freeze-thaw cycles and storage of the sera, could contribute to lower signal intensities as well; however, the Biotin/Strep combination is assumed to have the largest effect on this observation. The careful choice of secondary antibodies can hence affect the ability to characterize IgA and IgG antibodies in a common serum dilution in a quantifiable manner. 

The xMAP Intelliflex system used to Duplex the secondary antibodies comes as a ready-to-use system with an embedded computer and a touchscreen interface. Next to the advantage of the dual reporter system, the Intelliflex provides a much higher throughput with a read-time of approximately 20 min per 96-well plate, in comparison to approximately 45 min in the Luminex 200. In addition, the Intelliflex yields virtually identical signals compared to the Luminex 200 analyzer, which allows using the instrument for Singleplex assays.

As a strength of our study, we were able to quantify IgA and IgG antibodies in a comparably large and clinically well-characterized study of NPC cases and controls [10]. We were able to directly compare the performance of the assay with the previously published data using the same serum samples. Moreover, we used the same goat-anti-human anti-IgA and anti-IgG antibodies, either biotinylated or directly coupled to a fluorescent dye, to exclude variation introduced by differing properties of secondary antibodies obtained from multiple species or manufacturers. These antibodies were previously tested for their antibody-isotype specificity to prevent any cross-detection of IgA or IgG antibodies. 

One potential limitation of our assay is the generalizability of the simultaneous IgA and IgG antibody quantification to other assays or applications. While IgG antibodies are the most frequent antibody isotype in serum [23], IgA antibodies are more abundant in mucous secretions [24]. The quantitative detection of IgA and IgG antibodies in a common serum dilution is possible in NPC patients, where IgA antibodies in serum can reach levels as high as IgG antibodies (Figure 2A) but is more challenging in individuals with less strongly elevated IgA levels (Figure 2D). However, the characterization of two different antibody isotypes has been shown to be possible for other applications, e.g., the characterization of IgM and IgG antibodies to SARS-CoV-2 proteins [25] and eight antigens from five human pathogenic Borrelia species [26]. While IgM antibodies tend to be generally present at lower levels than IgG antibodies, they are high antibody-level markers of disease in the abovementioned applications. Hence, simultaneously detecting IgM and IgG antibodies at the same sample dilution is feasible in these applications. In contrast, simultaneous detection of IgG and the much less abundant serum IgA antibodies appears more challenging. Essentially, for every application, the assay conditions (secondary antibodies, serum dilution, etc.) have to be optimized individually based on the abundance of the analytes to ensure signals within the quantitative range.

A second limitation of our assay is the generalizability of the simultaneous IgA and IgG antibody quantification to other ethnicities and populations. Samples for assay development were derived from two Taiwanese studies [19]; however, no details about different ethnicities were reported (e.g., proportion of Chinese Han, indigenous Taiwanese, or Cantonese). The final evaluation of the Duplex assay was performed with serum samples from the UK-based Head and Neck 5000 study [10], of which 85% were of white ethnicity. Hence, performance characteristics across various demographic groups (sex, age, ethnicity, etc.) remain to be investigated in larger studies.

The developed Duplex assay can further be used in seroepidemiological studies to evaluate the use of an extensive EBV IgA and IgG marker panel for screening purposes. The implementation of novel screening methods is challenging, especially when screening for rare cancers such as NPC. However, within certain high-risk areas or high-risk populations (e.g., Taiwan), the implementation of screening is conceivable, and the improved cost-effectiveness of a duplexed IgA and IgG detection may contribute to the feasibility of NPC screening implementation based on several EBV IgA and IgG antibodies [27]. In addition, our development will likely advance future serological assays with multiple isotypes, such as Human Papillomavirus (HPV) antibody detection for oropharyngeal cancer screening and early detection [28], or for monitoring HPV vaccine response in urine samples [29]. 

## 5. Conclusions

In summary, we have successfully developed a Duplex serological multiplex assay for the simultaneous detection of IgA and IgG antibodies in the serum of NPC patients. The assay displays a promising tool for saving precious serum samples when evaluating IgA and IgG antibodies in seroepidemiological studies and, furthermore, provides a cost-optimized method for NPC screening within NPC-endemic areas.

## Figures and Tables

**Figure 1 cancers-15-02578-f001:**
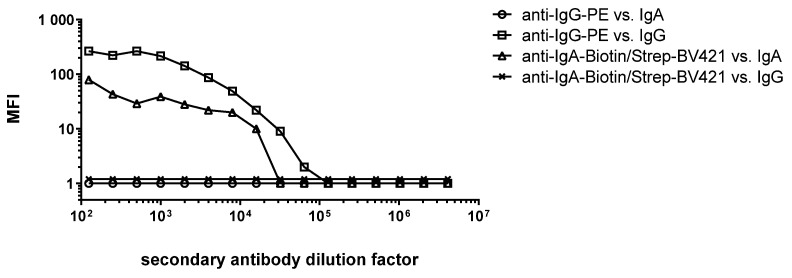
Titration of the final antibody-dye combination. The directly labeled anti-IgG-PE antibody, as well as the biotinylated anti-IgA antibody in combination with Strep-BV421, were tested against IgA and IgG from human serum to determine antibody specificity and optimal antibody concentration. MFI: median fluorescence intensity.

**Figure 2 cancers-15-02578-f002:**
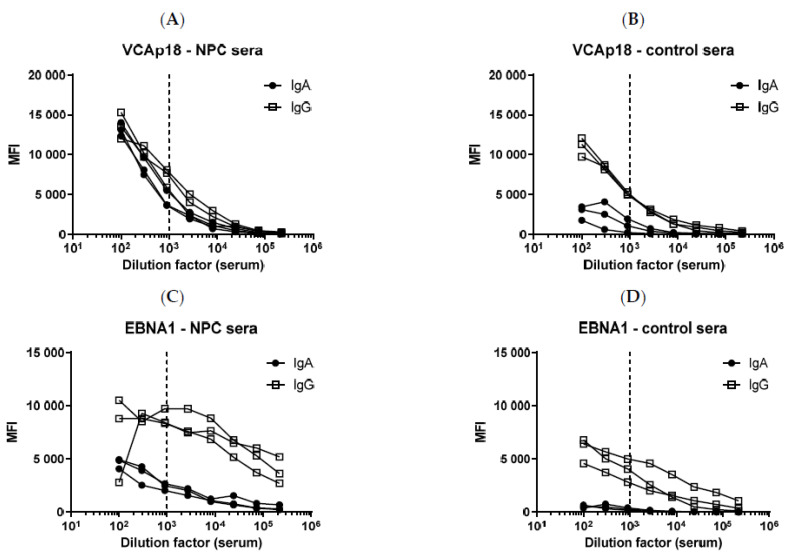
Serial dilution of sera from NPC patients and controls. Pooled sera were serially diluted to determine the optimal serum dilution factor for the simultaneous detection of EBV IgA and IgG antibodies. The dashed line indicates the serum dilution factor of 1:1000. IgA and IgG antibodies against the EBV antigens VCAp18 and EBNA1 are exemplarily shown for each NPC patient and control sera. (**A**): VCAp18—NPC sera. (**B**): VCAp18—control sera. (**C**): EBNA1—NPC sera. (**D**): EBNA1—control sera.

**Figure 3 cancers-15-02578-f003:**
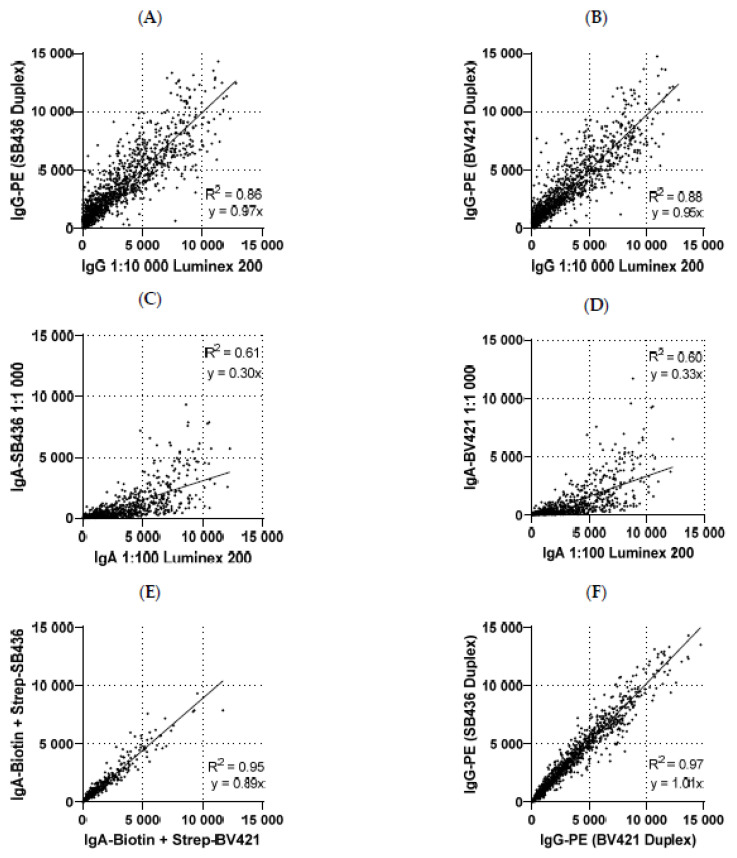
Comparison of IgA and IgG antibody signal intensities in Singleplex and Duplex assays with various reporter dyes. Data of all 13 analyzed antigens and 236 sera are summarized. (**A**,**B**): Signal intensities from anti-IgG-PE antibodies in a Duplex reaction with either SB436 or BV421 (1:1000 serum dilution) were compared to the signal intensity from anti-IgG-Biotin + Strep-PE measured in a 1:10,000 serum dilution in the Luminex 200. (**C**,**D**): Signal intensities from biotinylated anti-IgA antibodies in a Duplex reaction with either SB436 or BV421 (1:1000 serum dilution) were compared to the signal intensity from anti-IgA-Biotin + Strep-PE measured in a 1:100 serum dilution in the Luminex 200. (**E**): Correlation of anti-IgA-Biotin + BV421 and anti-IgA-Biotin + SB436 signals. (**F**): Correlation of the anti-IgG-PE signals in a Duplex assay with anti-IgA-Biotin + Strep-BV421 or anti-IgA-Biotin + Strep-SB436.

**Figure 4 cancers-15-02578-f004:**
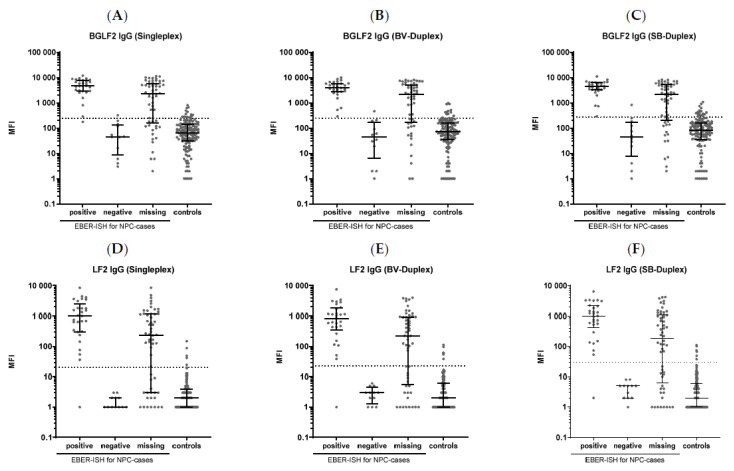
Comparison of antibody responses measured in a Singleplex Assay (Luminex 200) and a Duplex Assay (Intelliflex) stratified by EBER-ISH status for 97 NPC cases and 139 controls. Dashed lines indicate cut-offs based on ≥90% specificity among EBER-ISH negative NPC. (**A**): BGLF2 IgG (Singleplex). (**B**): BGLF2 IgG (Brilliant Violet 421 (BV)-Duplex). (**C**): BGLF2 IgG (Super Bright 436 (SB)-Duplex). (**D**): LF2 IgG (Singleplex). (**E**): LF2 IgG (BV-Duplex). (**F**): LF2 IgG (SB-Duplex).

**Table 1 cancers-15-02578-t001:** Overview of directly labeled and biotinylated antibodies and fluorescent dyes.

Product	Manufacturer	Product Number
**Directly labeled antibodies**		
goat anti-human IgA-PE	Dianova	109-115-011
goat anti-human IgG-PE	Dianova	109-115-098
goat anti-human IgA-DyLight405	Dianova	109-475-011
goat anti-human IgG-DyLight405	Dianova	109-475-098
**Biotinylated antibodies**		
biotinylated goat anti-human IgG	Dianova	109-065-098
biotinylated goat anti-human IgA	Dianova	109-065-011
Biotinylated mouse anti-tag from KT3 hybridoma supernatant [17]		
goat anti-mouse-Cy3	Dianova	115-165-146
**Dyes**		
BV421 Streptavidin Horizon	BD Biosciences	563259
Streptavidin-R-Phycoerythrin	Moss	SAPE-001
Streptavidin Super Bright 436 Conjugate	eBioscience	62-4317-82

**Table 2 cancers-15-02578-t002:** Cut-off values for LF2 and BGLF2 IgG antibodies according to EBER-ISH status (based on ROC analysis yielding ≥ 90% specificity) for 41 NPC tumors (29 EBER-ISH positive, 12 negative).

Antigen	Cut-Off			Sensitivity			Specificity		
	Singleplex	BV-Duplex	SB-Duplex	Singleplex	BV-Duplex	SB-Duplex	Singleplex	BV-Duplex	SB-Duplex
BGLF2 IgG	241	251	271	96.6	100	100	91.7	91.7	91.7
LF2 IgG	30 *	30 *	31	96.6	96.6	96.6	100	100	100

* Adjusted to confirm to the lower limit of quantification (LLOQ) of 30 MFI.

## Data Availability

The data presented in this study are available on request from the corresponding author. The data are not publicly available for data protection reasons.

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
