# Peer review of "Development of a Duplex Serological Multiplex Assay for the Simultaneous Detection of Epstein-Barr Virus IgA and IgG Antibodies in Nasopharyngeal Carcinoma Patients"

_cancers, 2023, doi:10.3390/cancers15092578_

Round 1

Reviewer 1 Report

This study by Simon et al. developed a duplex serological multiplex assay for the simultaneous detection of IgA and IgG antibodies in serum of NPC patients. By selection of appropriate secondary antibody/dye combinations and serum dilution that suitable for quantification of both IgA and IgG, the authors provided a method that save cost for NPC screening. The antigens panel was as described previously by the authors and collaborators (J Clin Microbiol. 2020; Clin Cancer Res. 2018),therefore the novelty and broad impact is considered to be relatively moderate, but the study is well carried out and the technique might be useful in NPC endemic regions. Some minor comments are:  

1. Detailed information regarding NPC patients and controls should be provided, including ethnics, sex and age. Are they Chinese Han, indigenous Taiwanese or Cantonese?

2. Page 3, “13 NPC-specific antigens”, ZEBRA or Zta is encoded by BZLF1 gene, it seems only 11 antigens, including one peptide of EBNA1 besides EBNA1 recombinant protein. Better described as 11 antigens/13 antibodies.

3. The full name of EBV genes should be spell out at least once in the manuscript. For example, EA-D is encoded by BMRF1, VCAp18 is encoded by BFRF3. Does IL2 correspond to BILF2?

4. The authors could discuss why IgG of some EBV antigens perform better than the cognate IgA, as NPC is an epithelial malignancy and the infection of EBV in NPC tissue is supposed to induce mucosal immunity.

Reviewer 2 Report

Comments are in the pdf
